# Application of phage display technology for the production of antibodies against *Streptococcus suis* serotype 2

Pattarawadee Sulong[1], Natsinee Anudit[2], Suphachai Nuanualsuwan[3,4], Segura Mariela[5], Kannika Khantasup[2,4,6] *

1 The Medical Microbiology Program, Graduate School, Chulalongkorn University, Bangkok, Thailand, 2 Department of Biochemistry and Microbiology, Faculty of Pharmaceutical Sciences, Chulalongkorn University, Bangkok, Thailand, 3 Department of Veterinary Public Health, Faculty of Veterinary Sciences, Chulalongkorn University, Bangkok, Thailand, 4 Food Risk Hub, Research Unit of Chulalongkorn University, Bangkok, Thailand, 5 Laboratory of Immunology, Faculty of Veterinary Medicine, University of Montreal, St-Hyacinthe, Quebec, Canada, 6 Vaccines and Therapeutic Proteins Research Group, the Special Task Force for Activating Research (STAR), Faculty of Pharmaceutical Sciences, Chulalongkorn University, Bangkok, Thailand

* Kannika.Kh@chula.ac.th

**Data Availability Statement:** All relevant data are within the manuscript and its Supporting Information files.

## Abstract

*Streptococcus suis* (*S. suis*) serotype 2 infection is a problem in the swine industry and responsible for most cases of human infection worldwide. Since current multiplex PCR cannot differentiate between serotypes 2 and 1/2, then serotype-specific antibodies (Abs) are required for serotype identification to confirm infection by serotype 2. This study aimed to generate Abs specific to *S. suis* serotype 2 by phage display from a human heavy chain variable domain (VH) antibody library. For biopanning, whole cells of *S. suis* serotype 2 were used as the target antigen. With increasing selection stringency, we could select the VH Abs that specifically bound to a *S. suis* serotype 2 surface antigen, which was identified as the capsular polysaccharide (CPS). From ELISA analysis, the specific phage clone 47B3 VH with the highest binding activity to *S. suis* serotype 2 was selected and shown to have no cross-reactivity with *S. suis* serotypes 1/2, 1, and 14 that shared a common epitope with serotype 2 and occasionally cause infections in human. Moreover, no cross-reactivity with other bacteria that can be found in septic blood specimens was also observed. Then, 47B3 VH was successfully expressed as soluble 47B3 VH in *E. coli* TG1. The soluble 47B3 VH crude extract was further tested for its binding ability in a dose-dependent ELISA assay. The results indicated that the activity of phage clone 47B3 was still retained even when the Ab occurred in the soluble form. A quellung reaction demonstrated that the soluble 47B3 VH Ab could show bioactivity by differentiation between *S. suis* serotypes 2 and 1/2. Thus, it will be beneficial to use this VH Ab in the diagnosis of disease or discrimination of *S. suis* serotypes Furthermore, the results described here could motivate the use of phage display VH platform to produce serotyping antibodies.

**Funding:** This research was funded by New Scholar Ratchadaphiseksomphot Endowment Fund from Chulalongkorn University, grant number DNS 60-095-33-006-1 and research fund from Faculty of Pharmaceutical Sciences, Chulalongkorn University, grant number Phar2559-RG02. This work also supported from CU Graduate School Thesis Grant.

**Competing interests:** The authors have declared that no competing interests.

## Introduction

*Streptococcus suis* (*S. suis*) is one of the most important swine pathogens and a zoonotic agent that can induce septicemia, deafness, meningitis, endocarditis, pneumonia, and arthritis in human [1, 2]. Based on the antigenic differences of capsular polysaccharides (CPS), it can be classified into 35 serotypes [3, 4], of which serotype 2 is considered to be the most prevalent and most pathogenic for both swine and humans worldwide [5]. The pathogenicity of serotype 2 may in part be due to its high invasiveness and many virulence factors, including CPS, extracellular protein factor, muramidase-released protein, suilysin, several adhesins, hyaluronate lyase, and surface antigen [6, 7]. In Thailand, *S. suis* infections of humans are more common in the northern region where eating raw pork and pig products is common [8, 9]. A review of the medical records revealed that infection by *S. suis* serotype 2 (96.4%) was predominant, but other serotypes could also cause disease in human, including serotypes 14 (4.5%), 24 (0.45%), 5 (0.3%), and 4 (0.15%) [10].

For microbiological diagnosis, an alpha-hemolytic reaction on blood agar plates with a Gram-positive cocci appearance can be further identified as *S. suis* by several biochemistry tests [11]. For serotype differentiation, serological typing with CPS-specific antibodies or multiplex PCR specific to the CPS gene has been routinely used. However, multiplex PCR cannot differentiate between serotypes 2 and 1/2 [12]. So, a serologic technique using CPS-specific antibodies (Abs) is still required as the standard procedure to confirm *S. suis* serotype 2, which is important for diagnosis and surveillance of emerging of *S. suis* infections.

Recently, phage display technology has been used widely to produce highly specific monoclonal antibodies (mAbs) in a short time, without animal use [13]. Based on antigen binding sites, the Ab fragments, such as the fragment of antigen binding (Fab), and single-chain variable fragments (scFv) have been constructed in the format of phage display libraries [14]. Moreover, the human heavy chain variable domain (VH), retaining antigen binding ability, has been developed [15]. With a molecular weight of around 15 kDa, VH Abs have many advantages over intact and large Ab fragments, such as a higher tissue penetration and the ability to target cryptic epitopes [16]. Because of their small size, VH Abs allow for easy large-scale production in a bacterial expression system [17]. For these reasons, VH Ab production based on phage display represents an alternative technology for the generation of diagnostic antibodies.

This study aimed to apply the phage display technology to produce mAbs specific to *S. suis* serotype 2 that could be used for discrimination between serotypes 2 and 1/2. The VH Abs were selected from the human VH antibody library. Biopanning against whole cells of *S. suis* serotype 2 was employed to enrich Ab binding to bacterial CPS. The binding ability and cross-reactivity of the selected VH Ab were tested using ELISA, while its bioactivity was also explored using the quellung reaction.

## Materials and methods

### Bacterial strains and growth conditions

*S. suis* serotype 2 reference strain ATCC 700794, serotype 1/2 reference strain NIAH 1318, serotype 1 reference strain NIAH 10227, and serotype 14 reference strain NIAH 13730 were kindly provided by Faculty of Veterinary Science, Chulalongkorn University, Bangkok, Thailand and the Faculty of Public Health, Kasetsart University Chalermphrakiat Sakon Nakhon Province Campus, Sakon Nakhon, Thailand. All *S. suis* cultures were grown in 5 mL of Todd-Hewitt broth (THB; Bacto™ Todd Hewitt Broth, Becton Dickinson, New Jersey, USA) for 18 h at 37˚C and then inoculated into 15 mL of fresh THB at 37˚C with shaking at 200 rpm until

the optical density at 600 nm wavelength ($OD_{600}$) reached 0.8. *Streptococcus pyogenes* reference strain DMS 3393 was grown in THB as described for *S. suis* above. *Staphylococcus aureus*, *Escherichia coli*, *Pseudomonas aeruginosa*, *Enterobacter aerogenes* were obtained from our laboratory collection. All bacteria were grown in Luria-Bertani (LB) (10 g of NaCl, 10 g of Tryptone 5 g of yeast extract per liter) medium and incubated at 37°C overnight and inoculated in 3 mL of LB at 37°C with shaking at 200 rpm until the OD600 reached 0.8.

## Phage biopanning

For the biopanning, the Human Domain Antibody Library (DAb) (Source BioScience, Nottingham, UK) with a diversity of $3 \times 10^9$ plaque-forming units (pfu) was used for this study. In whole cell preparation, the *S. suis* serotype 2 was confirmed to be well-encapsulated with CPS using the previously described cell surface hydrophobicity test [18] and then the cells were washed three times with phosphate buffered saline pH 7.4 (PBS) before use. The well-encapsulated whole cells of *S. suis* serotype 2 were blocked in PBS with 1% (w/v) bovine serum albumen (BSA) in PBS and incubated in a microcentrifuge tube (preblocked in 1%BSA in PBS for 1 h at room temperature) on a rotator for 1 h at room temperature. Meanwhile, non-specific phages were pre-absorbed in a microcentrifuge tube for 1 h at room temperature. Then, pre-absorbed phages at $8.8 \times 10^9$ pfu in 1% BSA in PBS were added to the tube containing $1 \times 10^9$ blocked *S. suis* cells. The mixture was rotated for an additional 1 h at room temperature. Unbound phages were removed by washing the cells five times with PBS containing 0.01% (v/v) Tween-20 (0.01% PBST) followed by five washes with PBS.

The stringency of selection was increased by decreasing the amount of *S. suis* serotype 2. In the first round, $1 \times 10^9$ cells of *S. suis* serotype 2 were used and the number of cells was reduced by two-fold in each subsequent round until the third round of biopanning. Moreover, the stringency of selection was increased by increasing the number of PBST washes in each round. The first round was washed with 0.01% PBST five times and then increased by one more wash each successive rounds. For the final washing step, the resuspended cells were moved to a fresh tube, preblocked with 1% BSA in PBS, as described above. After the final spin, the cell pellet was resuspended in 500 μL of 50 mM citrate buffer (pH 2.6) and rotated at room temperature for 5 min, to elute the bound phages. The tube was centrifuged (1,200x g, 4°C, 15 min) and the supernatant containing the eluted phages was transferred to a new microcentrifuge tube and neutralized with 200 μL 1 M Tris-HCl pH 8.0. Then, a final concentration of 1 mg/mL of trypsin (AppliChem, Darmstadt, Germany) (77 μL) was added to remove helper phage contamination. The titer of the eluted phage (output) was estimated, and an aliquot of the eluted fraction was used to infect *E. coli* TG1 cells for amplification to get input phages for the next round. The phage binding, elution, and amplification steps were performed for six rounds.

## Polyclonal phage ELISA

Polyclonal phage ELISA was performed to determine the effectiveness of *S. suis* serotype 2 specific phage enrichment. An overnight culture of *S. suis* serotype 2 ($1.5 \times 10^7$ cells/well) was coated overnight at 4°C. The plate was then washed five times with PBS. Non-specific binding was blocked with PBS containing 2% (w/v) powdered milk (MPBS) for 1 h at 37°C. After washing, input phages ($1.5 \times 10^8$ pfu in MPBS) were added and the plate was incubated at 37°C for 1 h, and then washed with 0.01% PBST. The cell bound phages were detected using a 1:2,000 dilution of anti-M13 horseradish peroxidase (HRP)-conjugate (Sino Biological, Wayne, USA) in MPBS and incubating for 1 h. Unbound antibodies were removed by washing with 0.01% PBST. The HRP activity was determined using TMB-substrate (Surmodics IVD,

Inc., Eden Prairie, USA) and monitoring the color change at 450 nm ($A_{450}$) using a CALIOstar Microplate reader (BMG LABTECH, Ortenberg, Germany).

## Monoclonal phage ELISA

To screen for positive phage clones that are specific for *S. suis* serotype 2, individual colonies from the sixth round of biopanning were picked and tested by ELISA. A single colony was inoculated in cell culture microplates and incubated at 37˚C with shaking at 200 rpm for 3 h. Then 4 x $10^8$ pfu of helper phage per well was added and incubated for 1 h at 37˚C. After incubation, the plates were spun at 2000 x g for 15 min and resuspended in 200 μL 2x tryptic soy broth (TYB) supplemented with 100 μg/mL ampicillin and 50 μg/mL kanamycin and incubated at 25˚C with shaking at 200 rpm overnight. The plates were then spun at 2000 x g for 15 min to harvest the amplified phages. The amplified phages in MPBS were added in ELISA well plates that were pre-coated with 1.5 x $10^8$ cells of *S. suis* serotype 2. Uncoated wells served as a negative control. After incubation, the plates were washed with 0.01% PBST. The cell bound phages were detected using a 1:2,000 dilution of anti-M13 HRP-conjugate (Sino Biological, Wayne, USA) as described above for the polyclonal phage ELISA. The positive clone was selected when the signal in the wells coated with *S. suis* serotype 2 was at least three-fold greater than the signal in the uncoated wells.

## Cross-reactivity

To determine the cross-reactivity of the positive phage, *S. suis* serotypes 1/2, 1, and 14, plus *Streptococcus pyoegnes*, *Staphylococcus aureus*, *Escherichia coli*, *Pseudomonas aeruginosa*, and *Enterobacter aerogenes* were coated for detection of cross-reactivity by ELISA. Bacterial cells were prepared as described above and coated at 4˚C overnight. The plate was washed with PBS. Non-specific binding was blocked with MPBS for 1 h at 37˚C. After washing with PBS, 50 μL of amplified phage were added and the plate was incubated at 37˚C for 1 h, then washed with 0.01% PBST. The cell-bound phages were detected using anti-M13 HRP-conjugate (Sino Biological, Wayne, USA), as described above for the polyclonal phage ELISA.

## Sequence analysis

Phagemids of the positive clones were extracted to confirm the presence of VH fragments in the recombinant phagemid DNA. The VH sequencing was performed using pR2-vector specific primers M13-rev: $5'-$ CAGGAAACAGCTATGAC $-3'$. The nucleotide sequences and the deduced amino acid sequences were compared with the Ab sequence in the GenBank sequence database.

## Target identification

To determine CPS specific VH Abs, different preparations were prepared as follows. Whole cells of *S. suis* serotype 2 were prepared as described above and then either left (untreated control), incubated at 95˚C for 30 min in coating buffer (heat-treated), or incubated in 20 μg proteinase K (ThermoFisher Scientific, CA, USA) at 37˚C for 1 h and then heated at 95˚C for 10 min (proteinase K-treated). Finally, a crude CPS extract (see below) of *S. suis* serotype 2 (5.45 μg/well) in 50 mM $NH_4HCO_3$ was also used as an Ab target.

The different preparations were coated overnight at 4˚C, while wells coated with 50 mM $NH_4HCO_3$ only served as the negative control. The plate was washed with PBS and non-specific binding was blocked with MPBS for 1 h at 37˚C. After washing with PBS, amplified phage in MPBS was added and the plate was incubated at 37˚C for 1 h, washed with 0.05% PBST, and

cell-bound phages were detected using anti-M13 HRP-conjugate (Sino Biological, Wayne, USA), as described above for the polyclonal phage ELISA.

## CPS extraction

The crude CPS extraction was prepared as previously described [19] except with some minor modifications. Briefly, *S. suis* serotype 2 was cultured in THB for 16 h at 37˚C. When the bacteria reached an $OD_{600}$ of 0.8, the cells were centrifuged, washed three times with phosphate-urea-magnesium sulphate (PUM) buffer, resuspended in PBS and chilled. The cell suspension was then autoclaved at 121˚C for 15 min and the crude CPS containing supernatant was recovered by centrifugation (15,000 x g, 4˚C, 15 min) and the pH confirmed to be within pH 7.0–8.0 and extracted in an equal volume of chloroform. The aqueous phase was harvested, supplemented to 25% (v/v) ethanol and 0.1 M $CaCl_2$ and incubated on ice for 15 min prior to centrifugation (15,000 x g, 4˚C, 15 min) and harvesting the supernatant. The concentration of ethanol in the supernatant was increased to 80% (v/v) and kept overnight at 4˚C to precipitate the CPS. The pellet containing CPS was collected by centrifugation, dissolved in 50 mM $NH_4HCO_3$ pH 7.8, dialyzed against $NH_4HCO_3$ pH 7.8 and then lyophilized. The lyophilized CPS was dissolved in $NH_4HCO_3$ pH 7.8 before use.

## Soluble VH expression in *E. coli*

For the production of the soluble VH Ab, the selected positive clone (47B3 VH) was used for expression. The phage clone was produced and grown at 37˚C in 100 mL of 2x TYB supplemented with 100 μg/mL ampicillin until the $OD_{600}$ reached 0.5. Isopropyl-1-thio-β-D-galactopyranoside (IPTG) at 0.5 mM was then added and incubated at 30˚C overnight to induce Ab expression. The next day, cells were harvested by centrifugation at 2600 x g for 20 min at 4˚C. The pellet was resuspended in 2.5 mL ice-cold periplasmic extraction solution [50 mM Tris/HCl, 20% (w/v) sucrose, 1 mM EDTA, pH 8] and incubated at 4˚C for 1 h with gentle mixing. The mixture was then centrifuged at 10,000 x g to collect the bacterial periplasmic fractions. The VH Ab content in the crude periplasmic extract was monitored visually following resolution through 8% sodium dodecylsulphate-polyacrylamide gel electrophoresis (SDS-PAGE) under reducing condition and staining with Coomassie blue. The *E. coli* TG1 without the phagemid vector was used as a negative control of expression. The soluble 47B3 VH was used as a crude extract in further experiments.

## Specificity of soluble VH

To determine the specificity of soluble 47B3 VH to *S. suis* serotype 2, an overnight culture of *S. suis* serotype 2 (1.5 x $10^8$ cells/well) and *S. suis* serotype 1/2 were coated in ELISA wells. The plate was washed five times with PBS. Non-specific binding was blocked with MPBS for 1 h at 37˚C. After washing five times with PBS, dilutions (1:2, 1:4 and 1:8) of soluble 47B3 VH crude extracts were added and the plate was incubated at 37˚C for 1 h and then washed five times with 0.01% PBST. The cell-bound Abs were detected using Protein A (1:1000) (Abcam, Cambridge, UK) in MPBS and further detected for HRP activity as described above for the polyclonal phage ELISA.

## Quellung reaction

The antigen-Ab reaction between soluble 47B3 VH and the CPS, causing the capsule to appear to swell, was observed by the quellung reaction. A sterile loop was used to take some colonies of a fresh overnight pure culture of *S. suis* serotype 2 or 1/2 and mixed into 100 μL of 0.85%

PBS. A drop of colony suspension (1.5 µL) was placed onto a slide and spread out. Then, a small drop (3.5 µL) of soluble 47B3 VH crude extract was placed on the first sections of a slide and spread out over the suspension. *S. suis* serotype 2 incubated with the commercial anti-*S. suis* serotype 2 pAb (Statens serum institut, Denmark) was used as a positive control. On the other hand, *S. suis* serotype 2 incubated with 0.85% PBS was used as a negative control. Methylene blue solution (5 µL) was added into the mixture, covered with a coverslip and incubated for 30 min before being observed under a microscope using an oil immersion lens.

### Statistical analysis

Data are expressed as the mean ± one standard deviation (SD). Comparisons were performed using an Unpaired t test for independent samples. Statistical analysis was performed using the SPSS version 22.0 software (SPSS Inc., Chicago, IL, USA). Statistical significance was accepted at the $p < 0.05$ level.

## Results

### Phage biopanning

For selection of phage-expressed VH Abs specific to *S. suis* serotype 2, a human VH antibody phage library with $8.8 \times 10^9$ pfu was used. The library was screened based on biopanning against whole cells of *S. suis* serotype 2. We prepared whole cells of well-encapsulated *S. suis* serotype 2 with a cell surface hydrophobicity that did not exceed 20%. To this end, preliminary work revealed that this was achieved by growing the *S. suis* serotype 2 culture to an $OD_{600}$ of 0.8–0.9 (data not shown). The progressive enrichment of the *S. suis* serotype 2-specific clones during the six successive biopanning rounds revealed that the enrichment increased about 47-fold from $1.73 \times 10^7$ pfu in the first round to $8.1 \times 10^8$ pfu in the sixth round (Table 1). This enrichment resulted in a significant increase in *S. suis* serotype 2 binding affinity of output phages in each round, as shown in Fig 1. These enriched phages showed no cross-reactivity with *S. suis* serotype 1/2, as tested in the polyclonal phage ELISA (Fig 1). Note that the polyclonal phage ELISA showed that the binding affinity of the phage did not increase anymore in the sixth round, indicating that the phage with maximum VH affinity was entirely enriched. Hence, we decided to finish the biopanning after the sixth round.

### Identification of positive phage clones by monoclonal phage ELISA

The antigen-specific phage clones were identified by monoclonal phage ELISA using the anti-M13 mAb (Fig 2). In total, 111 selected individual clones were randomly selected from the output phages of the sixth panning round and analyzed for their binding ability to *S. suis* serotype

**Table 1. Titer of input and output phage populations throughout six rounds of biopanning.**

| Biopanning | Phage input (pfu) | Phage output (pfu) |
|---|---|---|
| First round | $1.47 \times 10^{10}$ | $1.73 \times 10^7$ |
| Second round | $1.04 \times 10^{10}$ | $7.5 \times 10^8$ |
| Third round | $1.34 \times 10^{11}$ | $1.1 \times 10^8$ |
| Fourth round | $1.91 \times 10^{11}$ | $4.1 \times 10^8$ |
| Fifth round | $2.05 \times 10^{11}$ | $4.5 \times 10^8$ |
| Sixth round | $3.0 \times 10^{11}$ | $8.1 \times 10^8$ |
| Enrichment | | 46.8 |

pfu = phage-forming units.

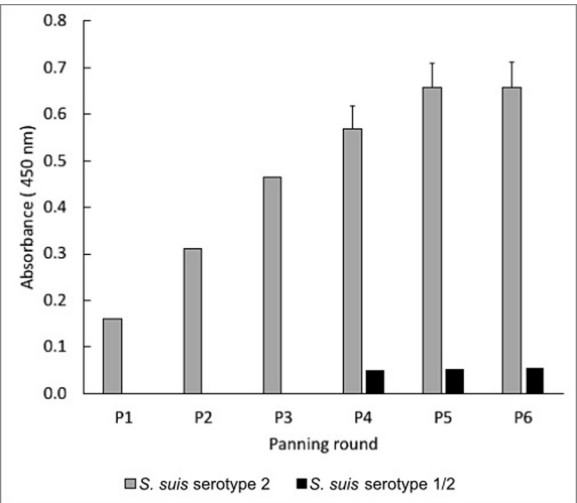

**Fig 1. Enrichment of phages specific to *S. suis* serotype 2, as determined by polyclonal phage ELISA.** Data are shown as the mean ± SD (n = 3).

2. Among the analyzed phage clones, six clones were considered positive, as in the signal seen in wells coated with *S. suis* serotype 2 was at least three-fold greater than the signal seen in the uncoated wells.

## Characterization of phage specific to CPS of S. suis serotype 2

The cross-reactivity profiles of the six positive clones (Fig 2) were determined against *S. suis* serotype 2, and other bacteria that could be the cause of false-positive results when testing samples suspected with *S. suis* serotype 2. The tested bacteria were divided into two groups. The first group was *S. suis* serotypes 1/2, 1, and 14, which are highly similar to serotype 2 cps structure (Fig 3) and have occasionally been reported from human cases. The second group was some of the bacteria that can be found in the bloodstream of sepsis patients, such as *Streptococcus pyogenes*, *Staphylococcus aureus*, *Escherichia coli*, *Pseudomonas aeruginosa*, and *Enterobacter aerogenes*. In the ELISA results, clone 47B3 and 36H1 showed no cross-reactivity with any of the tested bacteria, clone 20D9 had cross-reactivity with *S. suis* serotype 1, clones 68B5 and

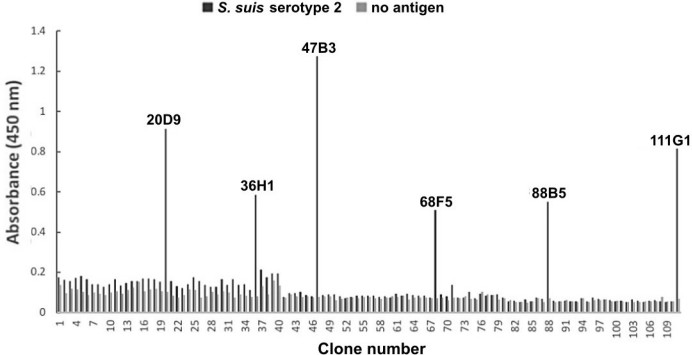

**Fig 2. Identification of monoclonal phages specific to *S. suis* serotype 2 after whole cell-biopanning using a monoclonal phage ELISA.**

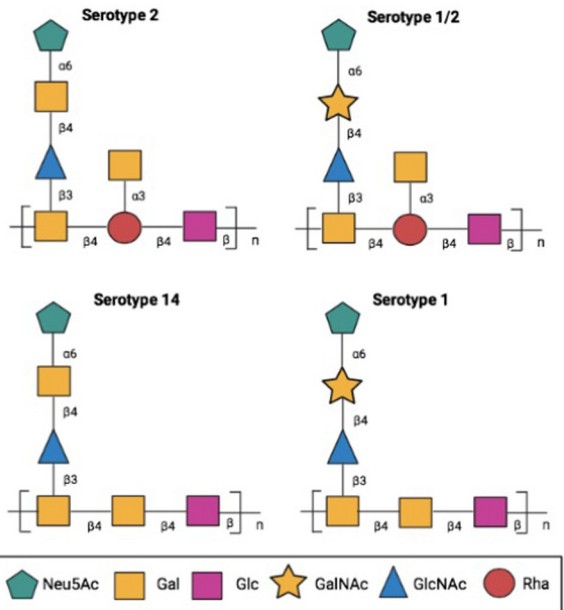

**Fig 3. The difference structure of the CPS repeating units among *S. suis* serotypes 2, 1/2, 1, and 14 as modified from Goyette-Desjardins et al. [20].** Abbreviations: *N*-acetyl-d-neuraminic acid (Neu5Ac), D-galactose (Gal), D-glucose (Glc), *N*-acetyl-d-galactosamine (GalNAc), *N*-acetyl-d-glucosamine (GlcNAc), and L-rhamnose (Rha).

88B5 had cross-reactivity with *Staphylococcus aureus*, and 111G1 had cross-reactivity with *Pseudomonas aeruginosa* (Fig 4 and Table 2).

The six positive phage clones were sequenced to determine the amino acid sequences of the framework and complementarity-determining regions (CDR). Multiple sequence alignment was used to estimate the sequence similarity. The multiple sequence alignment revealed that

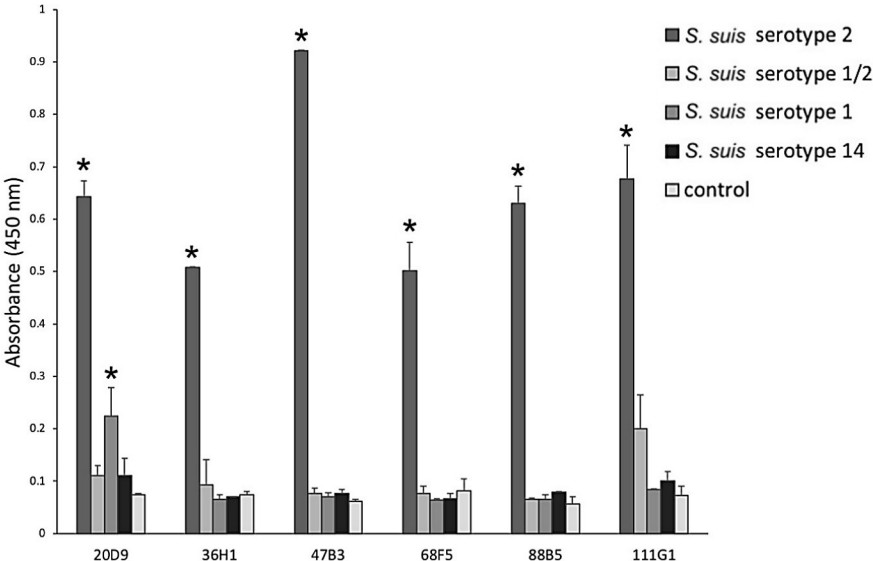

**Fig 4. Cross-reactivity of the six positive clones (20D9, 36H1, 47B3, 68F5, 88B5, and 111G1) against *S. suis* serotypes 2, 1/2, 1, and 14, as tested by ELISA.** Data are shown as the mean ± SD (n = 3). *$P < 0.05$ compared to the negative control.

**Table 2. Summary of the cross-reactivity of the six positive clones, as determined by ELISA.**

| Bacterial species | Strain | 20D9 | 36H1 | 47B3 | 68F5 | 88B5 | 111G1 |
|---|---|---|---|---|---|---|---|
| *Streptococcus suis* serotype 2 | ATCC 700795 | + | + | + | + | + | + |
| *Streptococcus suis* serotype 1/2 | NIAH 1318 | - | - | - | - | - | - |
| *Streptococcus suis* serotype 1 | NIAH 10227 | + | - | - | - | - | - |
| *Streptococcus suis* serotype 14 | NIAH 13730 | - | - | - | - | - | - |
| *Streptococcus pyogenes* | ATCC 19615 | - | - | - | - | - | - |
| *Staphylococcus aureus* | ATCC 25923 | - | - | - | + | + | - |
| *Escherichia coli* | ATCC 25922 | - | - | - | - | - | - |
| *Pseudomonas aeruginosa* | ATCC 27853 | - | - | - | - | - | + |
| *Enterobacter aerogenes* | ATCC 13048 | - | - | - | - | - | - |

**Notes**. (+) indicated significant differences between the experimental and control groups.

clones 68F5 and 88B5 were identical. Five of six clones had translational defects in their CDR 1, namely amber stop codons (TAG), as shown in Fig 5.

From the cross-reactivity profiles, 47B3 (47B3 VH) had the highest binding ability to *S. suis* serotype 2 and had no cross-reactivity with any tested bacteria, and so was selected for target identification and expression as a soluble VH in subsequent experiments.

## Target identification

Since biopanning was performed by means of whole cell-biopanning against well-encapsulated *S. suis* serotype 2, we assumed that 47B3 VH would bind to CPS, a surface exposed by phages. To test this hypothesis, we prepared different types of antigens based on CPS properties that are non-protein and heat stable, using the heat- and proteinase K-treated preparations to destroy protein antigens. Then, the different cell preparations were incubated with 47B3 VH and tested for target identification by ELISA. The 47B3 VH binding activity to the heat- and proteinase K-treated *S. suis* cells did not differ from that with the non-treated positive control, and the binding activity was increased in a dose-dependent manner (Fig 6). Since heat- and proteinase K-treatment does not denature CPS, the results could be interpreted as that the target on the cell surface may be CPS and not a protein. Moreover, to confirm that 47B3 VH was specific to CPS, the binding between the crude CPS extract of *S. suis* serotype 2 and 47B3 VH antibody was evaluated, where the 47B3 VH showed a similar binding specificity to CPS (Fig 7).

## Expression of the soluble VH

The selected positive clone 47B3 VH was used for expression of soluble VH. The sequencing data showed that 47B3 VH contained the amber codon at CDR1 (Fig 5). Actually, the amber stop codon is interpreted as a glutamine (CAG) residue in the TG1 amber suppressor strain and

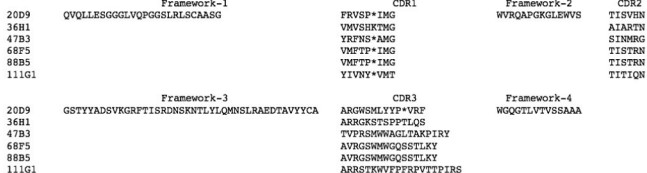

```
              Framework-1                      CDR1          Framework-2      CDR2
20D9    QVQLLESGGGLVQPGGSLRLSCAASG        FRVSP*IMG      WVRQAPGKGLEWVS    TISVHN
36H1                                      VMVSHKTMG                        AIARTN
47B3                                      YRFNS*AMG                        SINMRG
68F5                                      VMFTP*IMG                        TISTRN
88B5                                      VMFTP*IMG                        TISTRN
111G1                                     YIVNY*VMT                        TITIQN

              Framework-3                      CDR3          Framework-4
20D9    GSTYYADSVKGRFTISRDNSKNTLYLQMNSLRAEDTAVYYCA    ARGWSMLYYP*VRF    WGQGTLVTVSSAAA
36H1                                                  ARRGKSTSPPTLQS
47B3                                                  TVPRSMWWAGLTAKPIRY
68F5                                                  AVRGSWMWGQSSTLKY
88B5                                                  AVRGSWMWGQSSTLKY
111G1                                                 ARRSTKWVFPFRPVTTPIRS
```

**Fig 5. Amino acid sequence alignment of the six positive clones.** The CDRs and framework of the variable domains are indicated. The amber codon is marked by (*).

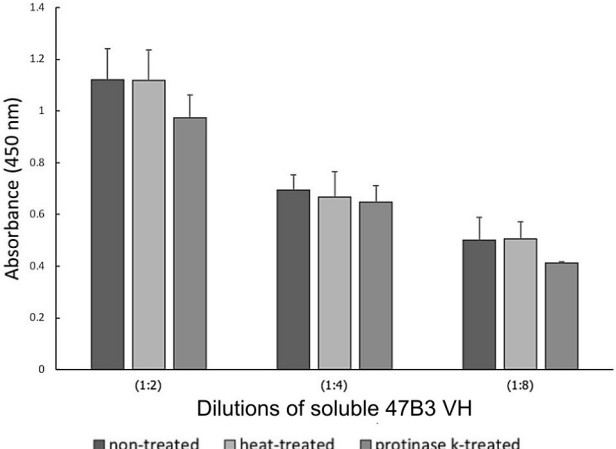

**Fig 6. Identification of the target antigen of 47B3 VH, as tested by ELISA.** Non-treated, heat-treated, and proteinase K-treated cells of *S. suis* serotype 2 were tested with different dilutions of 47B3 VH antibody. Data are shown as the mean ± 1SD (n = 3).

interpreted as a stop codon in HB2151, a non-amber suppressor strain. To get a complete soluble VH expression, *E. coli* TG1 was chosen as the expression host. After expression, the crude extract of soluble 47B3 VH in the periplasmic fraction was checked for the presence of VH by SDS-PAGE. Since phagemid in *E. coli* TG1 contained the 47B3 VH gene followed by the terminal c-Myc tag and the phage pIII protein, the soluble 47B3 VH was expressed as a fusion protein of approximately 58.7 kDa upon induction with IPTG (Fig 8) (VH 15 kDa, c-Myc tag 1.202 kDa, and pIII 42.5 kDa). By means of the pelB leader, we found that soluble 47B3 VH was expressed as the majority about 90% of the protein expressed in the bacterial periplasm, determined by SDS-PAGE. This evidence may imply the binding activity with serotype 2 in the next experiments would mostly come from our soluble VH. The yield of soluble 47B3 VH was approximately 0.43 mg per liter of culture as estimated from the band intensity of the BSA reference on SDS-PAGE.

## Characterization of soluble VH

Whether the soluble 47B3 VH still maintained its binding ability to *S. suis* serotype 2 after conversion into a soluble form was evaluated by ELISA. The results showed that the soluble 47B3

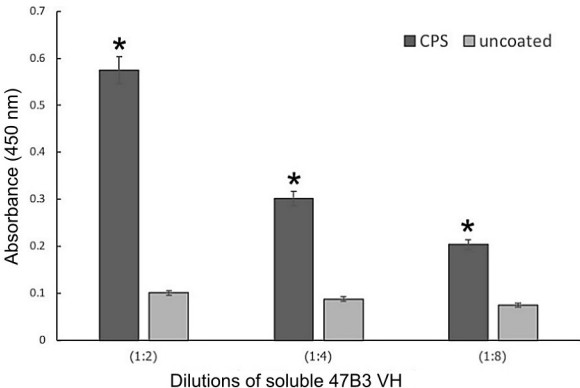

**Fig 7. Identification of the CPS binding ability of 47B3 VH at various dilutions, as tested by ELISA.** Data are shown as the mean ± 1SD (n = 3). *P < 0.05 compared to the negative control.

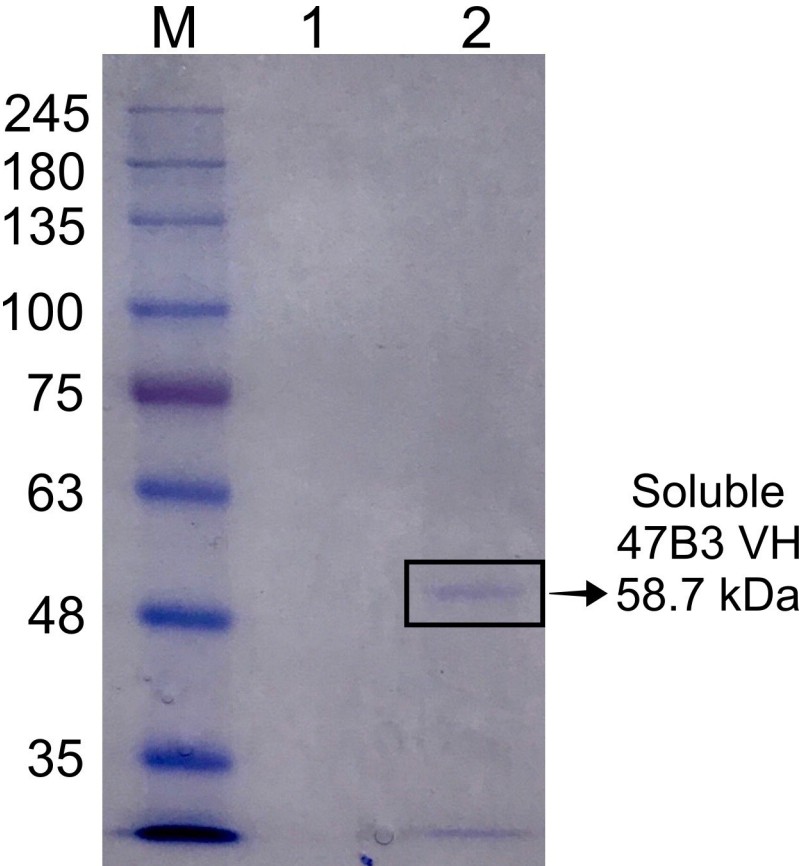

**Fig 8. Representative gel showing the soluble 47B3 VH expression, as revealed by SDS-PAGE with Coomassie blue staining.** M: Marker; 1: negative control (*E. coli* TG1 without phagemid vector); 2: soluble 47B3 VH. VH antibody production was separated on a reducing 8% SDS-PAGE.

VH had no cross-reactivity with serotype 1/2 and bound to *S. suis* serotype 2 in a specific and dose-dependent manner (Fig 9).

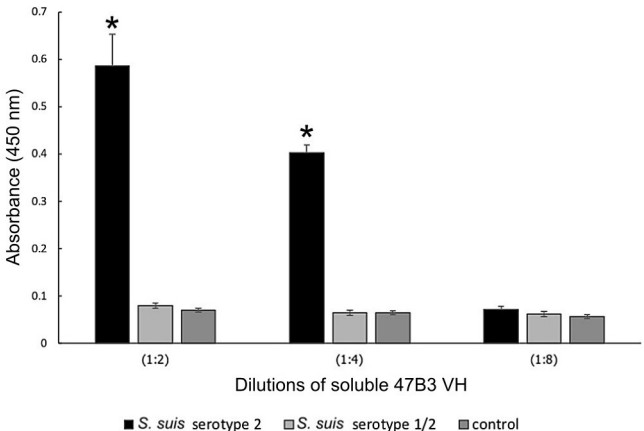

**Fig 9. Specificity of soluble 47B3 VH antibody to *S. suis* serotypes 2 and 1/2, as tested by ELISA.** Data are shown as the mean ± 1SD (n = 3). *$P < 0.05$ compared to the negative control.

In addition, the quellung test, which is the gold standard technique for serotyping Streptococcal bacteria, was used to test whether soluble 47B3 VH could be applied to differentiate between *S. suis* serotypes 2 and 1/2. The results showed that the commercial anti-*S. suis* serotype 2 pAb (positive control) and soluble 47B3 VH incubated with *S. suis* serotype 2 had positive results, with the capsule appearing as a sharply demarcated halo around the dark blue stained cell (Fig 10A and 10B). Meanwhile, serotype 1/2 incubated with soluble 47B3 VH (Fig 10C) and the negative control (*S. suis* serotype 2 without antibody) (Fig 10D) both showed a negative result with no clear and enlarged halo surrounding the stained cell. Therefore, the soluble 47B3 VH demonstrated a practical use to differentiate *S. suis* serotype 2.

## Discussion

Serotyping is one of the most important diagnostic methods for the surveillance and reporting of *S. suis* outbreaks as well as a guide for vaccine production. Molecular serotyping by multiplex PCR amplification of serotype specific *cps* genes has been an attractive tool as the assay can identify all serotypes except for between two pairs of serotypes: 1 and 14, and 2 and 1/2, as their serotype-specific genes share a high genetic similarity between them [2]. So, a serologic technique using CPS-specific Abs is still required as the standard procedure to confirm *S. suis* serotype 2. Serological typing with CPS-specific Abs is usually conducted using mAbs or pAbs, which need an animal and a long time in their production process. The current commercial diagnostic Abs for *S. suis* serological typing are pAbs, which have the additional drawback for serotyping of a high chance of cross-reactivity due to the recognition of multiple epitopes [21]. According to this problem, an additional step for pre-absorption with cross-reactive antigens before use is required [22]. Moreover, batch-to-batch variation in pAbs is inherent from the use of different animals. Therefore, the development of a method to produce a serotype-specific mAb that is fast, has no animal use, is easy to upscale, and offers batch-to-batch reproducibility is required. Over the past few years, phage display has emerged as a powerful technique for the production of mAb by means of biopanning. The whole cell-biopaning allows selecting mAb against cell surface targets in their native conformation [23–26]. For example, the production of a lipopolysaccharide-specific mAb using whole cells of *Legionella* as the antigen in biopanning [24]. From this finding, we had the hypothesis that, whole cell-biopanning could be used to obtain serotype-specific Abs, since CPS is the surface exposed layer to phage-expressed Abs. So, we started biopanning against well-encapsulated *S. suis* serotype 2 with a hydrophobicity of less than 20%. This whole cell-biopanning also bypasses the step for capsule purification that is complex and time consuming [27].

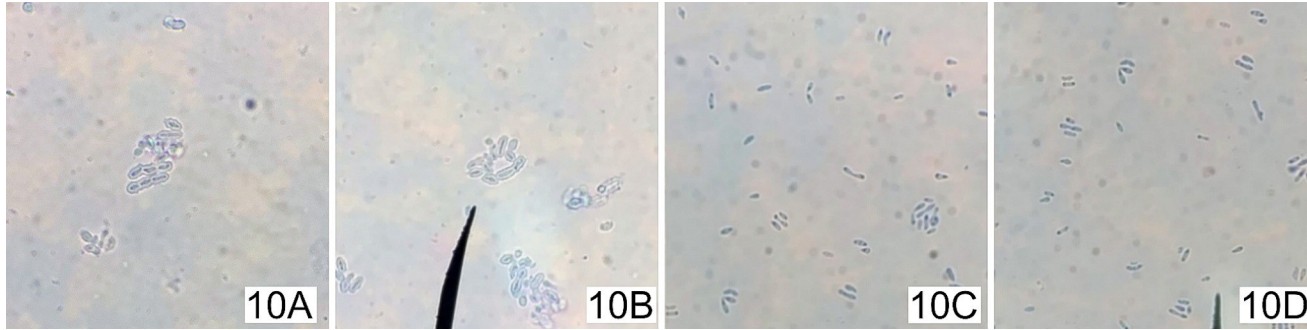

**Fig 10. Representative images of serotyping using the quellung test with soluble 47B3 VH.** Preparations of *S. suis* serotype 2 incubated with the commercial anti-*S. suis* serotype 2 pAb (A), *S. suis* serotype 2 incubated with soluble 47B3 VH (B), *S. suis* serotype 1/2 incubated with soluble 47B3 VH (C), and *S. suis* serotype 2 without antibody (D) were viewed under 1000X light microscope.

From the phage library, we succeeded in selecting a *S. suis* serotype 2-specific VH mAb, which recognized a heat stable and non-protein antigen that was verified to be CPS. A previous study reported a very low frequency of CPS-specific clones when attempting to produce mAbs directed against the CPS of *S. suis* serotype 2 by hybridoma technique. Around 3,000 clones were screened and only one clone demonstrated CPS binding, suggesting a non-immunogenic nature of CPS [20]. This non-immunogenic propriety of CPS could be overcome by selecting CPS-specific Abs by phage display technology, where the clear advantage is that screening fewer clones could be sufficient to get the positive phage clones. However, in addition, the smaller size of VH Abs compared to complete Abs may allow access to otherwise masked or cryptic antigenic epitopes (as discussed below).

Of six selected phage clones, the results revealed that phage clone 47B3 had the highest binding activity with *S. suis* serotype 2. Moreover, it showed no cross-reaction with *S. suis* serotypes 1/2, 1, and 14 that have been occasionally reported in human infections [28]. Meanwhile, although a number of mAbs directed against *S. suis* serotype 2 CPS have been reported, they showed cross-reaction between serotypes 2, 1/2, 1, and 14 [20]. In fact, the structures of *S. suis* CPS are formed by different arrangements of the monosaccharides into unique repeating units containing a side chain terminated by sialic acid (Fig 3) [29]. The cross reaction among serotypes 2, 1/2, 1, and 14 could be due to recognition of a part of the CPS repeated unit structures that are highly similar among serotypes, especially a common epitope at the sialic acid side chain [19, 20]. Notably, 47B3 VH showed the ability to discriminate between serotypes 2 and 1/2. The serotype 2 specify of 47B3 VH would come from its size being small enough to recognize a unique cryptic epitope of serotype 2 CPS. Similar discrimination between these serotypes was also reported by Goyette-Desjardins et al. [20]. In that study, hybridoma secreting mAb (16H11) was found to react with sialylated side chain of serotype 2. Since the only structural difference observed between serotypes 2 and 1/2 CPS was the galactose sugar bearing sialic acid in the side chains in serotypes 2, and *N*-acetylgalactosamine in serotype 1/2 (Fig 3), it is likely that this difference constituted an important unique epitope between serotypes 2 and 1/2. It was possible that 16H11 mAb and our 47B3 VH could bind to epitope containing this sugar bearing sialic acid. However, the precise epitope recognition of 47B3 VH needs to be further identified.

Moreover, the activity of 47B3 VH in phage form also had no cross-reactivity with bacteria that can be found in blood specimens for sepsis, such as *Streptococcus pyogenes*, *Staphylococcus aureus*, *Escherichia coli*, *Pseudomonas aeruginosa*, and *Enterobacter aerogenes* [30]. Focusing on encapsulated Gram-positive *S. pyogenes*, cross-reaction did not occur, since its CPS differs from *S. suis* by the presence of the hyaluronic acid polysaccharide [31]. It would be likely that, a unique arrangement of sugars in the repeated unit, the presence of terminal sialic acid, and the type of polysaccharides conferred a distinct antigenicity of this CPS.

Interestingly, serotype 2 specific 68B5 and 111G1 phages showed a weak binding affinity against *Staphylococcus aureus* and *Pseudomonas aeruginosa*, respectively. Since their CPS structure totally differs from *S. suis* serotype 2 [32, 33], for this reason, cross-reactivity may be caused by other partial similar epitopes on cell surface components that are shared between two bacterial strains.

Taken together, the cross-reaction data suggested 47B3 VH in phage form merited expression as a soluble form. Since 47B3 VH contained an amber codon in its CRD1, we decided to preliminary express 47B3 VH in *E. coli* TG1, an amber suppressor strain. The soluble 47B3 VH was expressed in *E. coli* TG1 and formed the protein in the crude extract of the bacterial periplasm. Note that, the soluble 47B3 VH could be enriched further if required in future studies using protein A or c-Myc tagged column chromatography.

The quellung reaction confirmed that soluble 47B3 VH showed bioactivity even when the binding site contained a single unpaired variable domain. Its binding affinity could be high

enough to induce capsular swelling, leading to the ability to differentiate between *S. suis* serotypes 2 and 1/2.

To date, the molecular tools have been developed to differentiate serotypes 2 from 1/2 such as PCR-restriction fragment length polymorphism assay (PCR-RFLP) [34], mismatch amplification mutation assay (MAMA)-PCR [35], and matrix-assisted laser desorption ionization time-of-flight mass spectrometry (MALDI-TOF MS) [36, 37]. However, these tools may not be available in low-resource settings or health stations. According to this limitation, antibody for serotype discrimination could be more practical with basic laboratory equipment.

Future studies will be required to investigate the 47B3 VH soluble expression in other amber non-suppressor *E. coli* hosts and to complete the protein enrichment to homogeneity. One suggestion is that the amber codon in CDR1 of 47B3 VH sequence could be synthetically substituted by glutamine and subcloned into expression plasmid to achieve a high level of cytoplasmic soluble protein in engineered *E. coli*, such as in the *SHuffle*® strain in order to produce large quantities of disulfide bond containing VH protein [38].

## Conclusion

This study showed that the novel VH mAb produced by phage display technology could specifically bind with *S. suis* serotype 2. Although all 29 serotypes were not tested, this VH Ab could differentiate between serotypes 2 and 1/2, which cannot be distinguished by PCR-based serotyping. Since the VH Ab can be produced easily using *E. coli* expression systems with fast cultivation and low production costs, it could be a promising tool for serotype discrimination of *S. suis* in order to complement the existing limitation of molecular serotyping. Furthermore, it has the potential for diagnosis of *S. suis* infectious disease, since serotype 2 is the most frequently reported serotype associated with human infections worldwide.

## Supporting information

**S1 Raw images. Raw images of gel data shown in Fig 8.**
(TIFF)

## Acknowledgments

The authors wish to thank Dr. Anusak Kerdsin, Faculty of Public Health, Kasetsart University Chalermphrakiat Sakon Nakhon Province Campus, Sakon Nakhon 47000, Thailand for providing *Streptpcoccus suis* reference strains.

## Author Contributions

**Conceptualization:** Pattarawadee Sulong, Kannika Khantasup.

**Data curation:** Pattarawadee Sulong, Kannika Khantasup.

**Formal analysis:** Pattarawadee Sulong, Natsinee Anudit.

**Funding acquisition:** Kannika Khantasup.

**Investigation:** Pattarawadee Sulong, Natsinee Anudit.

**Methodology:** Pattarawadee Sulong, Segura Mariela, Kannika Khantasup.

**Project administration:** Kannika Khantasup.

**Resources:** Suphachai Nuanualsuwan, Kannika Khantasup.

**Supervision:** Kannika Khantasup.

**Visualization:** Pattarawadee Sulong, Kannika Khantasup.

**Writing – original draft:** Pattarawadee Sulong, Kannika Khantasup.

**Writing – review & editing:** Kannika Khantasup.

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
