## [Decision Letter · Decision Letter 0]

27 May 2021

PONE-D-21-12605

Application of phage display technology for the production of antibodies against Streptococcus suis serotype 2

PLOS ONE

Dear Dr. Khantasup,

Thank you for submitting your manuscript to PLOS ONE. After careful consideration, we feel that it has merit but does not fully meet PLOS ONE’s publication criteria as it currently stands. Therefore, we invite you to submit a revised version of the manuscript that addresses the points raised during the review process.

A rebuttal letter that responds to each point raised by the reviewer(s). You should upload this letter as a separate file labeled 'Response to Reviewers'.A marked-up copy of your manuscript that highlights changes made to the original version. You should upload this as a separate file labeled 'Revised Manuscript with Track Changes'.An unmarked version of your revised paper without tracked changes. You should upload this as a separate file labeled 'Manuscript'.

We look forward to receiving your revised manuscript.

Kind regards,

Ellen R Goldman

Academic Editor

PLOS ONE

Journal Requirements:

3)  We suggest you thoroughly copyedit your manuscript for language usage, spelling, and grammar. If you do not know anyone who can help you do this, you may wish to consider employing a professional scientific editing service.  

4) We note that the grant information you provided in the ‘Funding Information’ and ‘Financial Disclosure’ sections do not match.

5) PLOS requires an ORCID iD for the corresponding author in Editorial Manager on papers submitted after December 6th, 2016. Please ensure that you have an ORCID iD and that it is validated in Editorial Manager. To do this, go to ‘Update my Information’ (in the upper left-hand corner of the main menu), and click on the Fetch/Validate link next to the ORCID field. This will take you to the ORCID site and allow you to create a new iD or authenticate a pre-existing iD in Editorial Manager. Please see the following video for instructions on linking an ORCID iD to your Editorial Manager account: https://www.youtube.com/watch?v=_xcclfuvtxQ

6) PLOS ONE now requires that authors provide the original uncropped and unadjusted images underlying all blot or gel results reported in a submission’s figures or Supporting Information files. This policy and the journal’s other requirements for blot/gel reporting and figure preparation are described in detail at https://journals.plos.org/plosone/s/figures#loc-blot-and-gel-reporting-requirements and https://journals.plos.org/plosone/s/figures#loc-preparing-figures-from-image-files. When you submit your revised manuscript, please ensure that your figures adhere fully to these guidelines and provide the original underlying images for all blot or gel data reported in your submission. See the following link for instructions on providing the original image data: https://journals.plos.org/plosone/s/figures#loc-original-images-for-blots-and-gels.

Reviewers' comments:

Reviewer's Responses to Questions

**Comments to the Author**

1. Is the manuscript technically sound, and do the data support the conclusions?

Reviewer #1: Yes

Reviewer #2: Yes

Reviewer #3: Yes

2. Has the statistical analysis been performed appropriately and rigorously? 

Reviewer #1: Yes

Reviewer #2: Yes

Reviewer #3: Yes

3. Have the authors made all data underlying the findings in their manuscript fully available?

Reviewer #1: Yes

Reviewer #2: Yes

Reviewer #3: Yes

4. Is the manuscript presented in an intelligible fashion and written in standard English?

Reviewer #1: Yes

Reviewer #2: Yes

Reviewer #3: Yes

5. Review Comments to the Author

Reviewer #1: This manuscript reports the development of antibodies for the detection and differentiation of Streptococcus suis serotype 2. The work appears to have been done in a complete and proper manner and should be published. I have only a few minor concerns that should be addressed.

1) An understanding of the serotypes of S. suis is essential for following this manuscript. The differences between types 2 and 1/2 (and also 1 and 14) are subtle variations in the polysaccharides. For the non-specialist, a fuller explanation should be given in the introduction, perhaps with a figure showing the differences.

2) Antibody 16H11 from ref. 25 also seems to differentiate between serotypes 2 and 1/2. This should be clearly stated in the manuscript and any relevant similarities and differences should be discussed.

3) Figure 4 is confusing, and seems to contain errors (at least the figure and the legend do not seem to agree). The legend states that identical residues are indicated by asterisks but this does not seem to be what the figure is showing. The figure uses a dash to indicate a stop codon. By convention, the asterisk is used to indicate a stop and the dash is used to indicate a gap in the sequence. I suggest using the asterisk for the stop and give the full sequence of each antibody without trying to indicate identical residues.

4) Overall the figures are of poor sharpness, even at the highest resolution available. As presented for this review, I think they are not of high enough quality for publication.

5) Line 24 indicates that some authors contributed equally to the work, but which authors is not stated.

Reviewer #2: General comments:

A better description of the human VH library that is being utilized along with a couple of references where other used it is warranted.

The main concern with this work is the soluble VH production. The plasmid is made for expression of the phage not a soluble protein, in the work see https://translational-medicine.biomedcentral.com/articles/10.1186/s12967-020-02538-y they moved the VH sequence to a pET-22b vector for efficient periplasmic expression. Thus, while you can produce some protein from this plasmid. It was not a system designed for good yields. This is compounded by the fact that all the work done with material was not with purified or quantified amounts. It would have been better to move your VH to a proper production plasmid, and produce and test properly purified material I think this would make this work stronger.

The x-axis for Figure 6 needs to be corrected.

Another minor issue was with the quellung test. It would have been nice to include a positive control, showing another antibody generate the same result, but it suffices as is. But again, a dose curve on the response would also have been a nice addition if using purified protein.

At one point I thought it would be an option to drop the soluble VH work and just publish the work with the VH expressed on the phage, are recent example of this using this library was published by Foods 2020, 9, 1230; doi:10.3390/foods9091230, but I do think have some data even if less than ideal is still a positive addition.

With an inclusion of more on the history and use of this library, and some possible other minor modifications, I believe this work should be able to be made acceptable for publication.

Reviewer #3: This is an interesting study and I enjoyed reviewing it.

Some comments

-The authors mentioned in different parts of the study that serotypes 2 and 1/2 cannot be differentiated by PCR and they must be tested by antisera. Indeed, this is no longer true since 2020: molecular tools to differentiate these two serotypes exist:

a) Matiasovic J, Zouharova M, Nedbalcova K, Kralova N, Matiaskova K, Simek B, Kucharovicova I, Gottschalk M. Resolution of Streptococcus suis Serotypes 1/2 versus 2 and 1 versus 14 by PCR-Restriction Fragment Length Polymorphism Method. J Clin Microbiol. 2020 Jun 24;58(7):e00480-20.

b) Lacouture S, Okura M, Takamatsu D, Corsaut L, Gottschalk M. Development of a mismatch amplification mutation assay to correctly serotype isolates of Streptococcus suis serotypes 1, 2, 1/2, and 14. J Vet Diagn Invest. 2020 May;32(3):490-494.

c) Scherrer S, Rademacher F, Spoerry Serrano N, Schrenzel J, Gottschalk M, Stephan R, Landolt P. Rapid high resolution melting assay to differentiate Streptococcus suis serotypes 2, 1/2, 1, and 14. Microbiologyopen. 2020 Apr;9(4):e995. doi: 10.1002/mbo3.995.

It is true that the antibody described in this study may be useful to labs not being equiped with PCR machines, so it is still intersting to publish these results.

-Lines 54-55: 6 serotypes (20, 22, 26, 32, 33 and 34) are no longer considered as S. suis

-I would have expected to find most phages that cross-react with serotypes 2 and 1/2...how the authors explain that most of them did not cross-react? chances to find the single epitope identifying the serotype 2 specific were very low...vs the finding of several phages that cross-react. I do really not understand how this happened...

-Soluble expression: do the authors know the yield?

-How the authors explain the reaction of some phages with S. aureus or P. aeruginosa? similar S. suis polysaccharides are present in such bacterial species?

6. PLOS authors have the option to publish the peer review history of their article (what does this mean?). If published, this will include your full peer review and any attached files.

Reviewer #1: No

Reviewer #2: No

Reviewer #3: No

---

## [Author Response · Author response to Decision Letter 0]

2 Jul 2021

Dear Editor-in-Chief, 

MS.Ref. No.: PONE-D-21-12605R1

Title: Application of phage display technology for the production of antibodies against Streptococcus suis serotype 2

We would like to submit a revised version of the manuscript for your consideration. Please find the response to the reviewers’ comments as follows.

Reviewer#1 Revisions responded to reviewer#1 were highlighted in blue, in revised manuscript. 

Reviewer’s comment 1: An understanding of the serotypes of S. suis is essential for following this manuscript. The differences between types 2 and 1/2 (and also 1 and 14) are subtle variations in the polysaccharides. For the non-specialist, a fuller explanation should be given in the introduction, perhaps with a figure showing the differences.

Response: Thank you for your suggestion. We have added Fig. 3 in Results part to demonstrate the difference between CPS of types 2 and 1/2 and also 1 and 14. The sentence “The first group was S. suis serotypes 1/2, 1, and 14, which have occasionally been reported from human cases” has been replaced by “The first group was S. suis serotypes 1/2, 1, and 14, which are highly similar to serotype 2 cps structure (Fig 3) and have occasionally been reported from human cases.” in Results part (line 315-317) of track changes revised version.

Fig 3. The difference structure of the CPS repeating units among S. suis serotypes 2, 1/2, 1, and 14 as modified from Goyette et al. [1]. Abbreviations: N-acetyl-d-neuraminic acid (Neu5Ac), D-galactose (Gal), D-glucose (Glc), N-acetyl-d-galactosamine (GalNAc), N-acetyl-d-glucosamine (GlcNAc), and L-rhamnose (Rha).

Reviewer’s comment 2: Antibody 16H11 from ref. 25 also seems to differentiate between serotypes 2 and 1/2. This should be clearly stated in the manuscript and any relevant similarities and differences should be discussed.

Response. You have raised an important point here. The sentences discussing about 47B3 VH binding epitope have been replaced by the sentences “Notably, 47B3 VH showed the ability to discriminate between serotypes 2 and 1/2. The serotype 2 specify of 47B3 VH would come from its size being small enough to recognize a unique cryptic epitope of serotype 2 CPS. Similar discrimination between these serotypes was also reported by Goyette et al [1]. In that study, hybridoma secreting mAb (16H11) was found to react with sialylated side chain of serotype 2. Since the only structural difference observed between serotypes 2 and 1/2 CPS was the galactose sugar bearing sialic acid in the side chains in serotypes 2, and N-acetylgalactosamine in serotype 1/2 (Fig 3), it is likely that this difference constituted an important unique epitope between serotypes 2 and 1/2. It was possible that 16H11 mAb and our 47B3 VH could bind to epitope containing this sugar bearing sialic acid. However, the precise epitope recognition of 47B3 VH needs to be further identified.” in the discussion part (line 505-516) of track changes revised version.

Reviewer’s comment 3: Figure 4 is confusing, and seems to contain errors (at least the figure and the legend do not seem to agree). The legend states that identical residues are indicated by asterisks, but this does not seem to be what the figure is showing. The figure uses a dash to indicate a stop codon. By convention, the asterisk is used to indicate a stop and the dash is used to indicate a gap in the sequence. I suggest using the asterisk for the stop and give the full sequence of each antibody without trying to indicate identical residues.

Response: Thank you for pointing this out. Due to patent concern, we would like to partially disclosure the VH amino acid sequence. In order to do that, we intently keep identical residues as asterisk. 

Reviewer’s comment 4: Overall the figures are of poor sharpness, even at the highest resolution available. As presented for this review, I think they are not of high enough quality for publication.

Response: Thank you for your suggestion. We have improved the quality of all figures in the revised manuscript. 

Reviewer#2 Revisions responded to reviewer#2 were highlighted in purple, in revised manuscript. 

Reviewer’s comment 1: The main concern with this work is the soluble VH production. The plasmid is made for expression of the phage not a soluble protein, in the work see https://translational-medicine.biomedcentral.com/articles/10.1186/s12967-020-02538-y they moved the VH sequence to a pET-22b vector for efficient periplasmic expression. Thus, while you can produce some protein from this plasmid. It was not a system designed for good yields. This is compounded by the fact that all the work done with material was not with purified or quantified amounts. It would have been better to move your VH to a proper production plasmid and produce and test properly purified material I think this would make this work stronger.

Response: Thank you for pointing this out. It would have been interesting to explore this matter. However, in this study, we aimed to demonstrate the use of phage display to develop VH and showed the discrimination between similar CPS antigens using obtained small VH antibody fragments. Therefore, experiments for convert VH sequence into pET plasmid series and expression as soluble will be tested in further study. However, the suggestion for expression strategy has been included in our manuscript as your suggestion. The sentences “One suggestion is that the amber codon in CDR1 of 47B3 VH could be synthetically substituted by glutamine to achieve a high level of cytoplasmic expression in E. coli, such as in the SHuffle® strains” have been replaced by “One suggestion is that the amber codon in CDR1 of 47B3 VH sequence could be synthetically substituted by glutamine and subcloned into expression plasmid to achieve a high level of cytoplasmic soluble protein in engineered E. coli, such as in the SHuffle® strain in order to produce large quantities of disulfide bond containing VH protein” in the discussion part line (554-558) of track changes revised manuscript. 

About VH purity aspect, we demonstrated that our soluble VH was the majority about 90% purity of the crude extract determined by SDS-PAGE results. The sentences “By means of the pelB leader, we found that soluble 47B3 VH was expressed as the majority of the protein expressed in the bacterial periplasm.” has been replaced by the sentences “By means of the pelB leader, we found that soluble 47B3 VH was expressed as the majority about 90% of the protein expressed in the bacterial periplasm, determined by SDS-PAGE. This evidence may imply the binding activity with serotype 2 in the next experiments would mostly come from our soluble VH. The yield of soluble 47B3 VH was approximately 0.43 mg per liter of culture as estimated from the band intensity of the BSA reference on SDS-PAGE” in the result part (line 395-401) of track changes revised version.

Reviewer’s comment 2: The x-axis for Figure 6 needs to be corrected.

Response: Thank you for your suggestion. We have changed the x-axis detail of figure. The Figure 6 are changed to Figure 7 in track changes revised manuscript.

Reviewer’s comment 3: Another minor issue was with the quellung test. It would have been nice to include a positive control, showing another antibody generate the same result, but it suffices as is. But again, a dose curve on the response would also have been a nice addition if using purified protein. 

Response: We have added the positive control for quellung test in Fig 10A to demonstrate that S. suis serotype 2 was incubated with the commercial anti-S. suis serotype 2 pAb. Moreover, the methods and results about quellung test using the commercial anti-S. suis serotype 2 pAb have been added in the materials and methods part (line 259-262) and the results part (line 423-430) of track changes revised version.

Reviewer’s comment 4: At one point I thought it would be an option to drop the soluble VH work and just publish the work with the VH expressed on the phage, are recent example of this using this library was published by Foods 2020, 9, 1230; doi:10.3390/foods9091230, but I do think have some data even if less than ideal is still a positive addition.

Response: Thank you for your suggestion. Compared with recommended work, they detected binding activities using recombinant phage expressing VH. However, our soluble VH was expressed using IPTG induced TG1 E. coli. TG1 containing 47B3 phagemid was induced and expressed the soluble form of 47B3 VH in its periplasm. The obtained soluble VH was subjected to test its binding activities as well. According to our earlier point, in this study, we aimed to demonstrate the use of phage display to develop VH and showed the discrimination between similar CPS antigens using obtained small VH antibody fragments. Therefore, experiments for soluble expression will be tested in further study.

Reviewer#3 Revisions responded to reviewer#3 were highlighted in green, in revised manuscript. 

Reviewer’s comment 1: The authors mentioned in different parts of the study that serotypes 2 and 1/2 cannot be differentiated by PCR and they must be tested by antisera. Indeed, this is no longer true since 2020: molecular tools to differentiate these two serotypes exist:

a) Matiasovic J, Zouharova M, Nedbalcova K, Kralova N, Matiaskova K, Simek B, Kucharovicova I, Gottschalk M. Resolution of Streptococcus suis Serotypes 1/2 versus 2 and 1 versus 14 by PCR-Restriction Fragment Length Polymorphism Method. J Clin Microbiol. 2020 Jun 24;58(7):e00480-20.

b) Lacouture S, Okura M, Takamatsu D, Corsaut L, Gottschalk M. Development of a mismatch amplification mutation assay to correctly serotype isolates of Streptococcus suis serotypes 1, 2, 1/2, and 14. J Vet Diagn Invest. 2020 May;32(3):490-494.

c) Scherrer S, Rademacher F, Spoerry Serrano N, Schrenzel J, Gottschalk M, Stephan R, Landolt P. Rapid high resolution melting assay to differentiate Streptococcus suis serotypes 2, 1/2, 1, and 14. Microbiologyopen. 2020 Apr;9(4):e995. doi: 10.1002/mbo3.995.

It is true that the antibody described in this study may be useful to labs not being equiped with PCR machines, so it is still intersting to publish these results.

Response: Thank you for your suggestion. We have added the sentence “To date, the molecular tools have been developed to differentiate serotypes 2 from 1/2 such as PCR-restriction fragment length polymorphism assay (PCR-RFLP) [2], mismatch amplification mutation assay (MAMA)-PCR [3], and matrix-assisted laser desorption ionization time-of-flight mass spectrometry (MALDI-TOF MS) [4, 5]. However, these tools may not be available in low-resource settings or health stations. According to this limitation, antibody for serotype discrimination could be more practical with basic laboratory equipment.” in the discussion part (line 542-548) of track changes revised version. 

Reviewer’s comment 2: Lines 54-55: 6 serotypes (20, 22, 26, 32, 33 and 34) are no longer considered as S. suis.

Response: We have changed the sentence from “it can be classified into 35 serotypes” to “it can be classified into 29 serotypes [6, 7]” in the introduction part (line 53) of track changes revised version. We have also changed the sentence from “Although all 35 serotypes were not tested, this VH Ab could differentiate between serotypes 2 and 1/2, which cannot be distinguished by PCR-based serotyping.” to “Although all 29 serotypes were not tested, this VH Ab could differentiate between serotypes 2 and 1/2, which cannot be distinguished by PCR-based serotyping.” in the conclusion part (line 564-566) of track changes revised version.

Reviewer’s comment 3: I would have expected to find most phages that cross-react with serotypes 2 and 1/2...how the authors explain that most of them did not cross-react? chances to find the single epitope identifying the serotype 2 specific were very low...vs the finding of several phages that cross-react. I do really not understand how this happened...

Response: May we give an assumption for this point. Most of antibody occurred the cross-reactivity between serotype 2 and 1/2 as shown in the study of Goyette et al. [1] But, after screening around 111 phages clones, our soluble 47B3 VH clone showed specifically bound only serotype 2. It could come from its size being small enough to recognize a unique cryptic epitope of serotype 2 CPS. It was possible that our 47B3 VH could bind to this sugar bearing sialic acid epitope or other cryptic epitopes in the CPS complex as well. We have described this hypothesis in the discussion part. However, in our opinion, if we screen more phage clones, we can get the chance to meet phages that cross-react with other subtypes. In our result, we got phage clone 20D9 that cross-reacted with type 1, one subtype that its CPS is similar with type 2.

Reviewer’s comment 4: Soluble expression: do the authors know the yield?

Response: The sentences described the soluble yield “The yield of soluble 47B3 VH was approximately 0.43 mg per liter of culture as estimated from the band intensity of the BSA reference on SDS-PAGE.” has been added in the result part (line 399-401) of track changes revised version. 

Reviewer’s comment 5: How the authors explain the reaction of some phages with S. aureus or P. aeruginosa? similar S. suis polysaccharides are present in such bacterial species?

Response: Thank you for pointing this out. The sentences “Interestingly, serotype 2 specific 68B5 and 111G1 phages showed a weak binding affinity against Staphylococcus aureus and Pseudomonas aeruginosa, respectively. Since their CPS structure totally differs from S. suis serotype 2 [8, 9], for this reason, cross-reactivity may be caused by other partial similar epitopes on cell surface components that are shared between two bacterial strains.” has been added in the discussion part (line 526-530) of track changes revised version.

Best regards

Kannika Khantasup, Ph.D.

Department of Biochemistry and Microbiology Faculty of Pharmaceutical Sciences Chulalongkorn University, Thailand

---

## [Editor Report · Decision Letter 1]

11 Oct 2021

Application of phage display technology for the production of antibodies against Streptococcus suis serotype 2

PONE-D-21-12605R1

Dear Dr. Khantasup,

We’re pleased to inform you that your manuscript has been judged scientifically suitable for publication and will be formally accepted for publication once it meets all outstanding technical requirements.

Kind regards,

Ellen R Goldman

Academic Editor

PLOS ONE
---

## [Editor Report · Acceptance letter]

15 Oct 2021

PONE-D-21-12605R1 

Application of phage display technology for the production of antibodies against *Streptococcus suis* serotype 2 

Dear Dr. Khantasup:

I'm pleased to inform you that your manuscript has been deemed suitable for publication in PLOS ONE. Congratulations! Your manuscript is now with our production department. 

Kind regards, 

on behalf of

Dr. Ellen R Goldman 

Academic Editor

PLOS ONE